# Development of Magnetocardiograph without Magnetically Shielded Room Using High-Detectivity TMR Sensors

**DOI:** 10.3390/s23020646

**Published:** 2023-01-06

**Authors:** Koshi Kurashima, Makoto Kataoka, Takafumi Nakano, Kosuke Fujiwara, Seiichi Kato, Takenobu Nakamura, Masaki Yuzawa, Masanori Masuda, Kakeru Ichimura, Shigeki Okatake, Yoshitaka Moriyasu, Kazuhiro Sugiyama, Mikihiko Oogane, Yasuo Ando, Seiji Kumagai, Hitoshi Matsuzaki, Hidenori Mochizuki

**Affiliations:** 1Device & Process Application Development Unit, Research & Development Center, Asahi Kasei Microdevices Corporation, Atsugi AXT Maintower 20F, 3050 Okata, Atsugi 243-0021, Kanagawa, Japan; 2Department of Applied Physics, Graduate School of Engineering, Tohoku University, 6-6-05 Aoba-yama, Aoba-ku, Sendai 980-8579, Miyagi, Japan; 3Spin Sensing Factory Corporation, Research Center for Rare Metal and Green Innovation, 403 468-1 Aramaki Aza-Aoba, Aoba-ku, Sendai 980-0845, Miyagi, Japan

**Keywords:** magnetocardiography, TMR sensors, signal space separation, projection operation, environmental noise

## Abstract

A magnetocardiograph that enables the clear observation of heart magnetic field mappings without magnetically shielded rooms at room temperatures has been successfully manufactured. Compared to widespread electrocardiographs, magnetocardiographs commonly have a higher spatial resolution, which is expected to lead to early diagnoses of ischemic heart disease and high diagnostic accuracy of ventricular arrhythmia, which involves the risk of sudden death. However, as the conventional superconducting quantum interference device (SQUID) magnetocardiographs require large magnetically shielded rooms and huge running costs to cool the SQUID sensors, magnetocardiography is still unfamiliar technology. Here, in order to achieve the heart field detectivity of 1.0 pT without magnetically shielded rooms and enough magnetocardiography accuracy, we aimed to improve the detectivity of tunneling magnetoresistance (TMR) sensors and to decrease the environmental and sensor noises with a mathematical algorithm. The magnetic detectivity of the TMR sensors was confirmed to be 14.1 pT_rms_ on average in the frequency band between 0.2 and 100 Hz in uncooled states, thanks to the original multilayer structure and the innovative pattern of free layers. By constructing a sensor array using 288 TMR sensors and applying the mathematical magnetic shield technology of signal space separation (SSS), we confirmed that SSS reduces the environmental magnetic noise by −73 dB, which overtakes the general triple magnetically shielded rooms. Moreover, applying digital processing that combined the signal average of heart magnetic fields for one minute and the projection operation, we succeeded in reducing the sensor noise by about −23 dB. The heart magnetic field resolution measured on a subject in a laboratory in an office building was 0.99 pT_rms_ and obtained magnetocardiograms and current arrow maps as clear as the SQUID magnetocardiograph does in the QRS and ST segments. Upon utilizing its superior spatial resolution, this magnetocardiograph has the potential to be an important tool for the early diagnosis of ischemic heart disease and the risk management of sudden death triggered by ventricular arrhythmia.

## 1. Introduction

Magnetocardiography, which detects heart magnetic fields outside the body, has a higher spatial resolution than electrocardiography, which measures the electric potential on the body surface produced by heart activities [1]. While electrocardiography needs an exercise stress on the subject for detecting ischemic heart disease, especially stable angina, it has been reported that magnetocardiography requires no stresses on the subject to diagnose them [2,3]. Magnetocardiography shows clear changes before and after percutaneous coronary intervention and coronary revascularization, whereas electrocardiography shows no significant differences [4,5]. The diagnostic accuracy of magnetocardiography on ischemic heart disease is reported to be as high as myocardial scintigraphy [6,7]. Moreover, relationships between the abnormality in the spatial mappings of heart magnetic fields obtained with magnetocardiography and Brugada syndrome [8], left intraventricular disorganized conduction in patients with dilated cardiomyopathy [9], right ventricular cardiomyopathy [10] and malignant arrhythmias [11] have been reported, attracting attention to magnetocardiography for the early detection of arrhythmias and ventricular fibrillation. These reports suggest that the magnetocardiography may play an important role in predicting ischemic heart disease, which is the highest cause of death in the world, as well as the ventricular arrhythmia, which triggers sudden death.

The only practicalized magnetocardiographs with superconducting quantum interference devices (SQUIDs) have not been widely used yet due to the huge running costs of liquid helium to cool the sensors down to ultra-low operating temperatures (−269 °C). As these magnetocardiographs need Dewar vessels with thick vacuum walls containing liquid helium, sensors are kept away from the body surface, which causes a limitation of the spatial detectivity [12]. Therefore, magnetocardiographs with small sensors that work at room temperatures are desired. As magnetic sensors are suitable for detecting minute magnetic fields such as heart magnetic fields, optically pumped magnetometers [13,14] and nitrogen-vacancy center magnetometers [15,16] have recently been attracting interest. However, as the former sensors have to be heated above 100 °C and the latter still remain at the research level, there are difficulties with magnetocardiography, which is an urgent issue. On the other hand, tunneling magnetoresistance (TMR) sensors are easily produced with a common process technique for integrated circuits and have been very widely used for commercial sensors, such as magnetic heads and current sensors, as they can be driven at room temperatures. Moreover, the smallness of the sensors and the needlessness of Dewar vessels essential for SQUIDs realize a high spatial resolution by placing the sensors close to the subjects’ body surfaces. In fact, the quadrupole has been observed in the magnetocardiogram of patients with Brugada syndrome [8], which suggests a higher spatial resolution brings a better diagnostic accuracy. In addition, it has been reported that brain magnetic fields, which are far smaller than heart magnetic fields, have been measured with scalp attached magnetoencephalography with TMR sensors [17]. Thus, TMR sensors, which are approachable to the signal source, are adequate for magnetocardiographs. 

For conventional magnetocardiographs, magnetically shielded rooms are essential to reduce environmental noise. While heart magnetic fields are up to several tens of pico teslas at the body surface, the terrestrial magnetic field is as high as 50 μT. There have been several reports in which environmental noises are reduced using magnetically shielded rooms when applying magneto-resistance sensors to the magnetocardiography [18,19]; however, the magnetically shielded rooms require special operation and high initial costs. Therefore, the development of a mathematical algorithm to establish a system without magnetically shielded rooms is desirable to spread the use of magnetocardiography. 

In order to measure the tiny heart magnetic fields, an algorithm to decrease the sensor noises down to 1 pT, which is lower than the signal, is also favorable. Although it has recently been reported that heart magnetic fields were observed with TMR sensors under an unshielded condition by the AC modulation and the impedance compensation [20], the detectivity was not enough for accurate diagnostics. As TMR sensors have higher sensitivity and higher 1/*f* noise in the low frequency domain compared to giant magnetoresistance (GMR) sensors [21], the suppression of sensor noises may be effective to improve the accuracy of magneto-cardiometry. 

The purpose of this work was to fabricate a magnetocardiograph that allows us to quickly obtain clear heart magnetic field maps without liquid helium and magnetically shielded rooms. According to previous studies, early diagnoses of ischemic heart disease [22], follow-up before and after coronary revascularizations [5] and prognosis prediction of dilated cardiomyopathy [9] are expected to be possible when the magnetic field detectivity reaches 1.0 pT_rms_. Here, we aimed to obtain the heart magnetic detectivity of 1.0 pT_rms_ in the frequency domain of 0.2–100 Hz with integration for 1 min. The issues to be solved in this work were the following:Improving the magnetic detectivity of TMR sensors with high sensitivity and low noises;The suppression of environmental noises with the algorithm;The suppression of sensor noises with the algorithm.

The target values of sensor noises and the suppression of environmental noises were 1.0 pT_rms_ and −60 dB, respectively. The latter was set to suppress the assumed environmental noises of 1 nT to be lower than 1 pT. We report that we successfully fabricated a magnetocardiograph whose heart magnetic field detectivity was 0.99 pT_rms_ in an office building under the unshielded condition. Moreover, we obtained magnetocardiograms and current arrow maps as clear as SQUID magnetocardiographs in the QRS and ST segments. These results may lead to the common use of high-accuracy magnetocardiographs in hospitals and clinics in the future. 

## 2. Materials and Methods

The thin films of TMR sensors were deposited on 5-inch Si/SiO_2_ (1 μm) wafers using sputtering apparatus (HC7200, Canon Anelva Corp., Kanagawa, Japan). TMR sensors were microfabricated using the common process technique for general integrated circuits. After the microfabrication, the wafers were first annealed at 325 °C in the external magnetic field of 1 T vertical to the sensing axis, followed by the second annealing at 225 °C with the field parallel to the sensing axis to rotate the easy axis of pinned layer by 90 degrees. By orthogonalizing the magnetic easy axes of the pinned and free layers, the transfer curve with the magnetic field along the hard axis of the free layers reflects the *M*-*H* curve for the hard direction of the free layers [23,24,25,26]. The first magnetic flux concentrators (MFCs) of permalloy were deposited to be 2.0 mm × 1.6 mm × 10 μm with the magnetic field of 100 mT vertical to the sensing axis. The chip size was 4.2 mm × 2.2 mm × 625 μm. As shown in Figure 1a,b, the second MFCs of permalloy were fixed to overlap the first MFCs with thermosetting resin. The sizes of the second MFCs were 2 mm × 20 mm × 800 μm. The transfer curve and the detectivity of the sensors were measured with the open-loop system in a triple magnetically shielded box. When constructing the closed-loop systems for magnetocardiography measurements, feedback coils were placed to cover each set of a TMR sensor and MFCs bonded on a printed circuit board, and they were mechanically fixed on the printed circuit board to avoid the physical interference between feedback coils, TMR sensors and MFCs. The sensor array consisted of using an acrylic frame with 96 sensor cells in which three sensors were vertically arranged. The subject of the magnetocardiography measurements was a male in his twenties with no history of heart disease in the past. During the measurements, the subject laid on his back to be covered with the sensor array on his chest on a table in a laboratory on the 18th floor of the 26-storied office building. After obtaining the data with 48 channels (288 sensors), the digital processing of signal space separation (SSS) (explained later), the common digital filter (the 50 Hz notch filter and the 100 Hz low-pass filter) and the signal average for one minute referencing the electrocardiograph signals took place.

## 3. Results

### 3.1. TMR Sensors

The stacking structure of Sample-A was Sub. Si, SiO_2_/Ta 5/Ru 20/Ta 5/Co_70.5_Fe_4.5_Si_15_B_10_ 100/Ru 10/Co_70.5_Fe_4.5_Si_15_B_10_ 100/Ru 0.4/Co_40_Fe_40_B_20_ 3/MgO/Co_40_Fe_40_B_20_ 3/Ru 0.9/Co_75_Fe_25_ 2/Ir_22_Mn_78_ 10/Ta 5 / Ru 10 (in nm). In order to reduce the variation in detectivity, we aimed to uniformize the film thickness of each layer by oblique sputtering. The material for the free layers was Co_70.5_Fe_4.5_Si_15_B_10_ with excellent flatness and a soft magnetic property [23,25,26], which has been reported to lead the small anisotropy field and high sensitivity. The free layers were microfabricated to be 600 × 82 μm with the longitudinal side along the sensing axis. The stacking structure of Sample-B prepared as a reference was Sub. Si, SiO_2_/Ta 5/Ru 20/Ta 5/Ni_80_Fe_20_ 130/Ru 0.9/Co_40_Fe_40_B_20_ 3/MgO/Co_40_Fe_40_B_20_ 3/Ru 0.85/Co_75_Fe_25_ 2/Ir_22_Mn_78_ 10/Ta 5/Ru 10. The films were deposited to have the incident sputtered particles perpendicular to the wafer. The material of the free layers was the most commonly used crystalline ferromagnetic material, Ni_80_Fe_20_ [24]. The free layers were microfabricated to be 100 × 140 μm with the short side along the sensing axis. Both Sample-A and Sample-B were set on the print circuit boards as shown in Figure 1a–c. Sample-A with the first and second MFCs showed transfer curves as shown in Figure 1d, reflecting the typical *M*-*H* curve for the hard direction of the free layers. High magnetic gain can be attained by placing the first MFCs to overlap the free layers but not the pinned layer nor the tunnel barrier layer. Figure 1e shows the detectivity spectra of Sample-A without MFCs, only with first MFCs and with the first and second MFCs measured with AC magnetic fields at 10 Hz. The amplitudes of the magnetic fields were 50 nT_p-p_ for the sample without MFCs and 1 nT_p-p_ with MFCs. One can see in the figure that two types of MFCs clearly improve the detectivity. In the sample with the first and second MFCs, the detectivity was 4.8 pT/Hz^1/2^ at 1 Hz and 11.8 pT_rms_ in the frequency band between 0.2 and 100 Hz, and the sensor noise at 1 Hz was 1714 nV/Hz^1/2^/V. The power line noise at 50 Hz and 1/*f* noise in a wide frequency range was reduced using the digital processing as shown below. Figure 1f shows the histogram of the detectivities of 108 sensors of Sample-A and 124 sensors of Sample-B. We obtained the average detectivity of 14.1 pT_rms_ and the standard deviation, *σ*, of 1.9 pT_rms_, which are much improved compared to Sample-B, whose averaged detectivity was 23.2 pT_rms_ and standard deviation was 16.4 pT_rms_.

### 3.2. Closed-Loop Systems

While the TMR sensors explained above have high sensitivities, the working range is as small as ten nano teslas, which leads to a saturation of output in the terrestrial magnetic field of about 50 micro teslas. Therefore, we constructed the closed-loop systems using feedback coils as shown in Figure 2a–c. By producing the magnetic field canceling the external field by introducing the current corresponding to the output of the bridge circuit with TMR sensors enhanced with the driving amplifier (OP295, Analog Devices Inc., Wilmington, MA, USA), the operation point can be fixed. The magnitude of the magnetic field applied to the sensor can be measured as the voltage of the resistance R_fb_, V_out_. One can also obtain a better linearity to the external field by using closed-loop systems with feedback coils fully surrounding the TMR sensors and the MFCs. With our configuration composed of TMR sensors and the closed-loop system, we obtained the working range of ±80 μT and the non-linearity in the magnetic field range of ±50 μT (roughly corresponding to the terrestrial field) of ±0.03%. By using this configuration, the ideal linear algebra was established among the magnetic vector data measured by the sensor array, which allowed us to apply the advanced digital processing such as SSS shown in the next section. 

### 3.3. SSS

Next, we verified a reduction in the environmental fields by SSS using the TMR sensor array configured as a closed-loop system. SSS is the method to divide the signal and the external environmental noises by the least-squares method using multiple sensor data [27]. As shown in Equation (1), the magnetic field vector, B(r), in the spherical coordinate can be expressed with functions derived from the internal space of the sensor array (the first term of the right side) and from the external space (the second term of the right side) with the spherical harmonics $Y_{l,m}\left(\theta ,\varphi \right)$.
(1)$$B\left(r\right)=-\mu \overset{L_{\mathrm{in}}}{\underset{l=1}{\sum }}\overset{l}{\underset{m=-l}{\sum }}\alpha_{l,m}\cdot \nabla \left(\frac{1}{{\left|r\right|}^{l+1}}\cdot Y_{l,m}\left(\theta ,\varphi \right)\right)\;-\;\mu \overset{L_{\mathrm{out}}}{\underset{l=1}{\sum }}\overset{l}{\underset{m=-l}{\sum }}\beta_{l,m}\cdot \nabla \left({\left|r\right|}^{l}\cdot Y_{l,m}\left(\theta ,\varphi \right)\right)$$

By putting Φ as the magnetic field measured by the sensor array, Φ can be written as a formula of matrix using $S_{\mathrm{in}}x_{\mathrm{in}}$ and $S_{\mathrm{out}}x_{\mathrm{out}}$ as follows:(2)$$\Phi ={\left[\begin{array}{cc} \begin{array}{cc} \phi \left(r_{1}\right) & \phi \left(r_{2}\right) \end{array} & \begin{array}{cc} \cdots & \phi \left(r_{N}\right) \end{array} \end{array}\right]}^{T}=\Phi_{\mathrm{in}}+\Phi_{\mathrm{out}}\;\Phi =S_{\mathrm{in}}x_{\mathrm{in}}+S_{\mathrm{out}}x_{\mathrm{out}}=\left[S_{\mathrm{in}}\;\;S_{\mathrm{out}}\right]\left[\begin{array}{c} x_{\mathrm{in}}\\ x_{\mathrm{out}} \end{array}\right]$$
where $S_{\mathrm{in}}$ corresponds to the spherical harmonics of the internal signal in Equation (1), $S_{\mathrm{out}}$ is the spherical harmonics of the external signal and $x_{\mathrm{in}}$ and $x_{\mathrm{out}}$ are the coefficients of α and β, respectively. $S_{\mathrm{in}}$ and $S_{\mathrm{out}}$ are the base vectors that can be calculated with the positions and sensitivities of the sensors. Here, by estimating $x_{\mathrm{in}}$ and $x_{\mathrm{out}}$ as $\hat{x_{\mathrm{in}}}$ and $\hat{x_{\mathrm{out}}}$ using the least-squares method using measured Φ and base vectors $S_{\mathrm{in}}$ and $S_{\mathrm{out}}$, Φ can be divided into the estimated value of the internal signal $\hat{\Phi_{\mathrm{in}}}$ and the external signal $\hat{\Phi_{\mathrm{out}}}$.
(3)$$\hat{x}=\left[\begin{array}{c} \hat{x_{\mathrm{in}}}\\ \hat{x_{\mathrm{out}}} \end{array}\right]={\left(S^{t}S\right)}^{-1}S^{t}\Phi$$
(4)$$\hat{\Phi }=\left[\hat{\Phi_{\mathrm{in}}}+\hat{\Phi_{\mathrm{out}}}\right]=S\hat{x}=\left[S_{\mathrm{in}}\;\;S_{\mathrm{out}}\right]\left[\begin{array}{c} \hat{x_{\mathrm{in}}}\\ \hat{x_{\mathrm{out}}} \end{array}\right]$$
(5)$$\hat{\Phi_{\mathrm{in}}}=S_{\mathrm{in}}\hat{x_{\mathrm{in}}}$$
(6)$$\hat{\Phi_{\mathrm{out}}}=S_{\mathrm{out}}\hat{x_{\mathrm{out}}}$$
where $\hat{\Phi }$ is used as the heart signal in the magnetocardiography measurements.

The experimental result of the environmental noise reduction by the SSS is shown below. The sensor array was constructed using 96 sensor cells containing three TMR sensors in each, and the TMR sensors were arranged perpendicularly, as shown in Figure 3a. As can be seen from the upper side, 48 channels comprised the double layer. By three-dimensionally surrounding the subject’s chest with the arched array through the underarms, the noise reduction effect by SSS can be increased. Figure 3b shows the block diagram of the measuring system. After detecting the V_out_, as explained above, the computer received the data through an RC high-pass filter (0.2 Hz) to eliminate the terrestrial magnetic noises, amplifier (AD8676, Analog Devices Inc., Wilmington, MA, USA), RC low-pass filter (194 kHz) for anti-aliasing, A/D converter (ADS1296, Texas Instruments Inc., Dallas, TX, USA), the FPGA and the memories, followed by the signal processing, such as SSS. The measured result of the noise reduction with this array configuration is shown in Figure 3c. In a regular laboratory in a 26-storied office building, we observed the magnetic field of 100 nT_p-p_ at 7 Hz produced by a Helmholtz coil three meters away from the sensor array, simulating the environmental magnetic field. By inputting the simulated magnetic field with the known frequency, amplitude and direction, the reducing effect of the environmental magnetic field by SSS could be evaluated. In the spectra before reducing the noise (Raw data), one can see peaks corresponding to the environmental noises in the low frequency domain due to elevators and so on, as well as ones around 20 Hz and 50 Hz due to the experimental equipment and the air conditioning in the laboratory, other than the input signal from the coil at 7 Hz. In the spectra after applying SSS (After SSS), both the signal at 7 Hz and the environmental noises around 20 Hz and 50 Hz are decreased. Comparing the mean values at 7 Hz along the *Y*-axis before and after applying SSS, the reduction effect of the environmental noises was estimated to be −73 dB, achieving the target value of −60 dB. Here, the calculation condition of the SSS was set to be $L_{\mathrm{in}}$ = 3 and $L_{\mathrm{out}}$ = 4 in Equation (1).

The magnetocardiography measurement with the sensor array is depicted in Figure 4a. The SSS calculation was performed with $L_{\mathrm{in}}$ = 3 and $L_{\mathrm{out}}$ = 4 in Equation (1), and the data for one minute (50 heart beats) were used for the signal average. The waveforms before and after each digital processing are shown in Figure 4b: (i) shows the time series waveform of the *Z*-sensor in the lower layer of Channel 27 (ch27), where the environmental noises of sub-nano tesla around 20 Hz due to the air conditioning and ones of several nano teslas at 50 Hz due to the commercial electric power are super-positioned; (ii) shows the time series data after applying the SSS and the digital filter, where the environmental noises are almost removed and R waves buried in the environmental noises are observed; and (iii) shows the smoothened waveform of 50 heart beats by the signal average for one minute, where the QRS and T waves are clearly observed.

### 3.4. Projection operation

Further reduction in the sensor noise was performed using the projection operation. The projection matrix used for the projection operation is calculated based on the method of cross validation. Cross validation is a calculation algorithm with the repetition of the analysis with a part of the dataset divided into *k* groups and a validity check with the rest. In this work, we divided the data from the sensor array into two parts (*k* = 2) and calculated the projection matrix by computing the base vector for each group. Figure 5a shows the flowchart of the projection operation. Adding in calculating Φ with all the data from the whole sensor array, the digital processing from the SSS to the signal average was separately performed with data from channels with even numbers, $\Phi_{A}$, and with odd numbers, $\Phi_{B}$, to obtain $\hat{\Phi_{\mathrm{in}\_A}}$ and $\hat{\Phi_{\mathrm{in}\_B}}$. In addition, the data of unused channels in the calculations of $\hat{\Phi_{\mathrm{in}\_A}}$ and $\hat{\Phi_{\mathrm{in}\_B}}$ were estimated by computing $S_{in}\hat{x_{\mathrm{in}}}$ corresponding to each channel using $\hat{x_{\mathrm{in}}}$ obtained in Equation (3). Here, we performed a singular value decomposition on $\hat{\Phi_{\mathrm{in}\_A}}$ and $\hat{\Phi_{\mathrm{in}\_B}}$ to obtain the base vectors $L_{1}$ and $L_{2}$ in the time domain (or spatial domain), followed by calculating the projection matrices $P_{1}\left(=L_{1}\cdot L_{1}^{T}\right)$ and $P_{2}\left(=L_{2}\cdot L_{2}^{T}\right)$. Here, $P_{1}$ and $P_{2}$ are projection matrices to the space consisting of the signals and noises of selected sensors. Comparing the $\hat{\Phi_{\mathrm{in}}}$ calculated using data from all channels, the projection operation to $P_{1}$ and $P_{2}$ enabled us to reduce the sensor noises by retaining the amplitude of the signals (i.e., the projection to $P_{1}$ reduced the noises derived from the channels with odd numbers and $P_{2}$ did the same from the channels with even numbers compared to $\hat{\Phi_{\mathrm{in}}}$).

Figure 5b shows the time domain data of the *Z*-sensor in the lower layer of ch27. These measurements were carried out without the subject to compare the sensor noises before and after the projection operation. While *σ* = 1.77 pT_rms_ before the projection operation, it decreased to 0.99 pT_rms_ after the projection operation, which is below the target value of the sensor detectivity 1.0 pT_rms_. It was also confirmed that the noise reduction effect was −4.5 dB by comparing the second norms of all the sensors. 

Figure 5c shows the heart magnetic field maps obtained with the *Z*-sensors of the lower layer close to the body surface in 8 × 6 channels measured through the flowchart shown in Figure 5a. It can be seen that the polarity of the heart signal reverses near the heart, suggesting that the heart magnetic field is observed. The signals of R waves can be observed with the sensors away from the heart, including ones placed on the side of the body. These results mean that heart magnetic fields are observed from three-dimensional directions, which is expected to improve the accuracy of the current source estimation compared to measurements only from the front.

### 3.5. Preparing the Magnetocardiogram

To create detailed current arrow maps upon the heart, the dataset was reconstructed onto the divided area of 200 × 200 mm at 40 mm above the body surface with 9 × 9 segments at 25 mm intervals. Once $\hat{x_{\mathrm{in}}}$ was calculated by SSS, the magnetic field distribution on an imaginary sensor array surface, different from the real positions and angles of the sensors, could be easily estimated by preparing vector $S_{\mathrm{in}}$ for $\hat{x_{\mathrm{in}}}$ relating to the positions and angles to be calculated. Therefore, the magnetocardiogram can be optimized for diagnostic application as required. Figure 6a shows the magnetocardiogram reconstructed by estimating the *Z*-axis magnetic field, which was processed through the same procedure shown in Figure 5a except that $S_{\mathrm{in}}$ was adjusted in the SSS calculation. 

Figure 6b shows the magnetic field strength maps in contour lines at peaks of R and T waves along the *Z*-axis and the current arrow map calculated from the positional differentiation along the *Z*-axis. The current arrow maps were calculated by $I_{x}=\Delta B_{z}/\Delta y$ and $I_{y}=-\Delta B_{z}/\Delta x$. At the R and T waves, it can be seen that the current passes to the lower left of the heart, which is similar to the pattern observed in a healthy subject [10]. Figure 7a shows the time variation of the current arrow map in the QRS domain. The calculation conditions of SSS were $L_{\mathrm{in}}$ = 3 and $L_{\mathrm{out}}$ = 4 and $L_{\mathrm{in}}$ = 6 and $L_{\mathrm{out}}$ = 4. Although $L_{\mathrm{in}}$ = 6 gave slightly more detailed information in the domain near the S wave, both conditions resulted in an extremely clear magnetocardiogram comparing favorably with a measurement by a SQUID performed in magnetically shielded rooms (Figure 7b) in the QRS and ST domains. In the time variation during the QRS domain, the magnetocardiogram is showing the excitation of the ventricular septum (17 ms), leading to the excitation of the left ventricle (40 ms) and current flow to the right ventricle (65 ms). In these QRS and ST domains, relationships have been reported between the abnormality in the magnetocardiogram maps and heart disease, such as ischemic heart disease [6,7] and ventricular arrhythmia [8,9,10]. 

The spatial resolution can be improved by increasing $L_{\mathrm{in}}$, which is important for pattern analysis on unhealthy subjects. In this work, the measurements up to $L_{\mathrm{in}}$ = 6 allowed us to observe the detailed distribution on a healthy subject. However, we still need to collect further experimental data to distinguish how large $L_{\mathrm{in}}$ needs to be, but we believe this system can be applied to disease determination, such as in a previous report [28], by setting an appropriate $L_{\mathrm{in}}$.

## 4. Discussion

### 4.1. TMR Sensors

During the thin film deposition in this work, the incident angle of sputtering particles was set to be about 39 degrees onto the wafer surface to minimize the in-plane distribution of film thickness to reduce the variation of resistivities and the sensitivities of sensors taken out from a wafer. In this configuration, the variation of the detectivities of Sample-A was greatly reduced compared to that of Sample-B deposited by facing target type sputtering. In general, oblique sputtering deposition with crystalline targets is known to lead the growth of columnar crystals on wafers, leading to the deterioration of the flatness. When the flatness of the tunnel barrier layer in the TMR thin films is worsened, the electronic 1/*f* noise is increased due to the charge trapping in the tunnel barrier layer and at the interfaces of the tunnel junctions [29]. Moreover, the degrading of the flatness leads to magnetization domain hopping between the metastable ripple states, causing the enhanced magnetic 1/*f* noise [30]. Figure A1a,b in Appendix A shows cross-sectional images observed by scanning transmission electron microscope (STEM) on the TMR thin films with Ni_80_Fe_20_ free layers deposited by facing target type sputtering and oblique sputtering, respectively. In both images, characteristic shadows of columnar crystals in Ni_80_Fe_20_ can be seen. The grain sizes of those crystals are increased in the films deposited by oblique sputtering, which causes the deteriorated flatness of the MgO tunnel barrier layer. Here, to inhibit the development of crystallization in the free layers, amorphous Co_70.5_Fe_4.5_Si_15_B_10_ was selected as the material for the free layers. As a result, the columnar crystals were significantly reduced, and the flatness of MgO was greatly improved. A previous work [26] reported that the flatness of Co_70.5_Fe_4.5_Si_15_B_10_ with a thickness of 30 nm showed a similar value to that of Ni_80_Fe_20_. The difference between the previous report [26] and our work may be explained by the presence or absence of the development of columnar crystals being emphasized in our work because the free layer was as thick as 130 nm. The relationships between the average surface roughness, R_a_, measured by the atomic force microscope (AFM) and the magnetic and electronic noises fabricated in this work are shown in Figure A1d,e, respectively. The shape of the free layers in each sample is the same as in Sample-B. Here, the electronic 1/*f* noise was measured with the external magnetic field along the sensing axis to saturate the sensors, and the magnetic 1/*f* noise showed the value obtained by eliminating the effect of electronic noise from that measured at the center of the operation range in the magneto-resistive curve. As expected, the magnetic noise decreased with the R_a_. On the other hand, the electronic noise did not show the simple behavior against R_a_. In samples deposited by oblique sputtering, the use of Co_70.5_Fe_4.5_Si_15_B_10_ resulted in a lower R_a_ and electronic noise. However, in the samples with Ni_80_Fe_20_ free layers the facing target type sputtering induced improved electronic noise compared to oblique sputtering. This may be caused by the composition ratio, the crystal orientation or the defect density in the MgO layers, although the cause is not clear at present. 

Next, we discuss the shape of the free layers to obtain higher sensitivity. In general, the diamagnetic field, which occurs with the magnetic field in ferromagnetic materials, is decreased as the material is longer along the field [31]. Therefore, it was expected that a lower diamagnetic field and a higher sensitivity would be achieved by stretching the free layer along the sensing axis. However, when the free layer contained a single Co_70.5_Fe_4.5_Si_15_B_10_ sheet, the magnetic domain structure in the Co_70.5_Fe_4.5_Si_15_B_10_ layer appeared to set the vector sum of the magnetic moment in the free layer to zero. Figure A2a shows the image observed with a Kerr microscope in a multilayer structure of Sub. Si, SiO_2_/Ta 5/Ru 20/Ta 5/Co_70.5_Fe_4.5_Si_15_B_10_ 170/Ru 0.4/Co_40_Fe_40_B_20_ 3/Ta 2.5 patterned to be 100 × 140 μm. The triangle magnetic domain structure can be seen at the bottom of the pattern, which shows the development of a closure domain. As the direction of the magnetic easy axis was parallel or antiparallel to the sensing axis in the triangle domain, the hysteresis of the magneto-resistive curve was developed when this domain was directly under the magnetic tunnel junctions, decreasing the sensitivity near the zero-field state. When stretching the free layer along the sensing axis, the triangle domain expanded in the free layer, leading to a decrease in the sensitivity. Therefore, the magnetic domains must be unified to obtain long free layers along the sensing axis. In this work, we aimed to eliminate the magnetic domain structure by sandwiching the non-magnetic Ru film with double Co_70.5_Fe_4.5_Si_15_B_10_ sheets in the free layer to magnetostatically couple the Co_70.5_Fe_4.5_Si_15_B_10_ sheets on the side of the patterns. The result of magnetic domain observation on a pattern with a multilayer structure of Sub. Si, SiO_2_/Ta 5/Ru 20/Ta 5/Co_70.5_Fe_4.5_Si_15_B_10_ 100/Ru 10/Co_70.5_Fe_4.5_Si_15_B_10_ 100/Ru 0.4/Co_40_Fe_40_B_20_/Ta 2.5 is shown in Figure A2b. In this case, the whole pattern had a single magnetic domain. By using this stacking structure and microfabricating the free layer to be very long along the sensing axis, the sensitivity of Sample-A in the frequency band between 0.2 and 100 Hz reached 327 μV/nT/V on average, which is improved compared to 116 μV/nT/V in Sample-B. 

### 4.2. Environmental Noise

For the magnetocardiography measurements without magnetically shielded rooms, the largest problem is the noises, such as the terrestrial magnetic field and city noise. Particularly, city noise drastically interrupts the heart field as its amplitude is in the order of nano teslas in the same frequency band as heart beats. Therefore, magnetic field shielding by software is crucially important. Gradiometers are widely used to reduce noise disturbance [17,20]. While SSS requires a large number of sensors for the calculation process according to the degree of calculation ($L_{\mathrm{in}}$, $L_{\mathrm{out}}$), a gradiometer has advantages in that it needs only two sensors, and the configuration of the system is simple and easy to handle. However, a gradiometer is difficult to use with environmental noise where a spatial gradient exists. In addition, as the gradiometer is a method to obtain the difference in outputs between different sensors, it decreases not only the environmental noise but also the signal itself. As the noises have no correlation between sensors, the noise of the gradiometer is enhanced after the subtraction. 

In this work, we succeeded in decreasing the environmental noise by −73 dB using the software shield of SSS and without sacrificing the detectivity. General single magnetically shielded rooms decrease the environmental noise by about −20 dB, which means our SSS has the effect of more than a triple magnetically shielded room.

Performing environmental noise reduction in SSS depends on the configuration and error of sensors. We assume there are two reasons why we could obtain the large noise reduction effect. The first is the use of three-axis sensors that three-dimensionally surrounded the body through the underarms, leading to an increase in the information of the spatial magnetic field distribution and to the improvement of the calculation accuracy with spherical harmonics used for the base vectors. Let us compare the condition number that indicates the accuracy of the estimated solution by computing the ratio of the maximum and minimum singular values under a condition of $L_{\mathrm{in}}$ = 6 and $L_{\mathrm{out}}$ = 4. In the configuration where the three-axis sensors were three-dimensionally placed to the side of the body, the condition number was calculated to be 1226. When the sensors on the side of the body were lifted upward to form the flattened array shape, the condition number increased to 9849, which is worse than that of the original configuration. In addition, when all sensors in the array were replaced with *Z*-axis sensors, the condition number was 2305, worse than that with three-axis sensors.

The second reason is the adoption of a closed-loop system with feedback coils surrounding the whole TMR sensor configuration to increase the linearity against the input magnetic field, which improved the accuracy of the sensors enough to apply the advanced digital signal processing. It is known that the sensor errors affect the reduction in environmental noise [27]. In the array we constructed in this work, we estimated that the environmental noise reduction effect would be −66.8 dB and −61.3 dB when the non-linearity was 0.1% and 0.3%, respectively. We presumed that we could obtain the almost expected noise reduction effect by improving the sensor non-linearity down to about 0.03%. 

Referencing the power spectral density in Figure 3c, the environmental noise of about 10 nT at 50 Hz derived from the commercial electric power will not be problematic as it can be eliminated with the notch filter. The purpose of SSS is to reduce the noises around 20 Hz and the environmental 1/*f* noise. These noises are originally less than 1 nT and decrease below 1 pT after reducing by −73 dB using SSS. Moreover, they decrease to about 0.1 pT after the signal average for one minute, which is one digit smaller than the sensor noise. Although there were a number of noise sources, such as elevators and air conditioning around the measurement location, we used no countermeasures, such as placing the measurement system away from the noise sources. We think that there is almost no limitation to the installation location even without magnetically shielded rooms. Additionally, the heart magnetic signal and the environmental noises from the external space were separately obtained by SSS. Therefore, it is possible to avoid the disturbance by suspending the measurement when sudden environmental noises occur or eliminating the measurement data while the environmental noises exceed the default value by monitoring the external field.

### 4.3. Sensor Noise

While TMR sensors were used in this work, GMR sensors are also known as magnetic sensors with high detectivities operatable at room temperatures. In general, while TMR sensors have a higher magneto-resistance ratio and sensitivity than GMR sensors, the electronic 1/*f* noise that occurs in the insulating MgO barrier in the low frequency band is larger than with GMR sensors [21]. Nevertheless, sensors with higher sensitivities hide the noises of the following circuit such as amplifiers and resistive elements. Therefore, even if the detectivities are the same in TMR and GMR sensors, the use of TMR sensors is expected to result in a lower final noise as a module after digital processing than when using GMR sensors. In the magnetocardiograph fabricated in this work, we adopted electronic devices with low noises, and the system noise of the circuit was estimated to be 1.5 pT_rms_ between 0.2 and 100 Hz as a designed value before the signal average, which is thought to hardly have any impact compared to the noise of TMR sensors (14.1 pT_rms_). 

Finally, we discuss the noise reduction in SSS based on the detectivity of the TMR sensors and the final detectivity after digital processing, even though the direct estimation was difficult as the environmental noise was dominant before SSS. The original detectivity of the TMR sensors, 14.1 pT_rms_, was decreased to 0.99 pT_rms_, by −23.0 dB, through the digital processing of the SSS, signal average and projection operation. The noise reduction effect by the signal average for one minute (50 heart beats) was estimated to be −17.0 dB, and the SSS and projection operation were estimated to reduce the noise by −6.0 dB. Adding to the −4.5 dB at the projection operation part, the SSS also decreased the noise by about −1.5 dB. A part of the sensor noise, as well as the environmental noise, was thought to be cut away to the external space, which helped to achieve the target detectivity of 1.0 pT. 

## 5. Conclusions

In this work, we succeeded in fabricating a high-accuracy magnetocardiograph without magnetically shielded rooms by improving the TMR sensors and decreasing environmental noises and sensor noises with digital processing. TMR sensors with a high detectivity of 14.1 pT_rms_ were prepared by applying a new multilayer structure and the shape of free layers. The large 1/*f* noise in TMR sensors was reduced by selecting the amorphous Co_70.5_Fe_4.5_Si_15_B_10_ known to have the excellent flatness. The variation of the detectivities was suppressed by oblique sputtering. By adopting SSS, the reduction effect of the environmental noise attained −73 dB, which overtakes triple magnetically shielded rooms. Moreover, the digital processing combining SSS and the projection operation led to a reduction in the sensor noise by −6.0 dB. As the result, the detectivity of the magnetocardiograph without a magnetically shielded room reached 1.0 pT_rms_ with the integration for one minute in the frequency band of 0.2–100 Hz. By using this system, we obtained a remarkably fine magnetocardiogram and current arrow maps in the QRS and ST segments, comparing favorably with a measurement by a conventional SQUID performed in magnetically shielded rooms. As the clear magnetocardiogram was obtained without magnetically shielded rooms at room temperature, the future task is to examine the availability of this magnetocardiograph through clinical trials. Although the P wave due to atrial depolarization was not clearly observed because of the shortage of the detectivity with the signal average for one minute, it may be possible to detect it by further improvements of the TMR sensors and digital processing. The spatial resolution and heart field detectivity can be improved by placing sensors closer to the heart by reforming the shape of the frame of the array and the setting of the sensors, as the distance between the sensors in the sensor array shown in this work and the body surface was approximately 40 mm. 

## Figures and Tables

**Figure 1 sensors-23-00646-f001:**
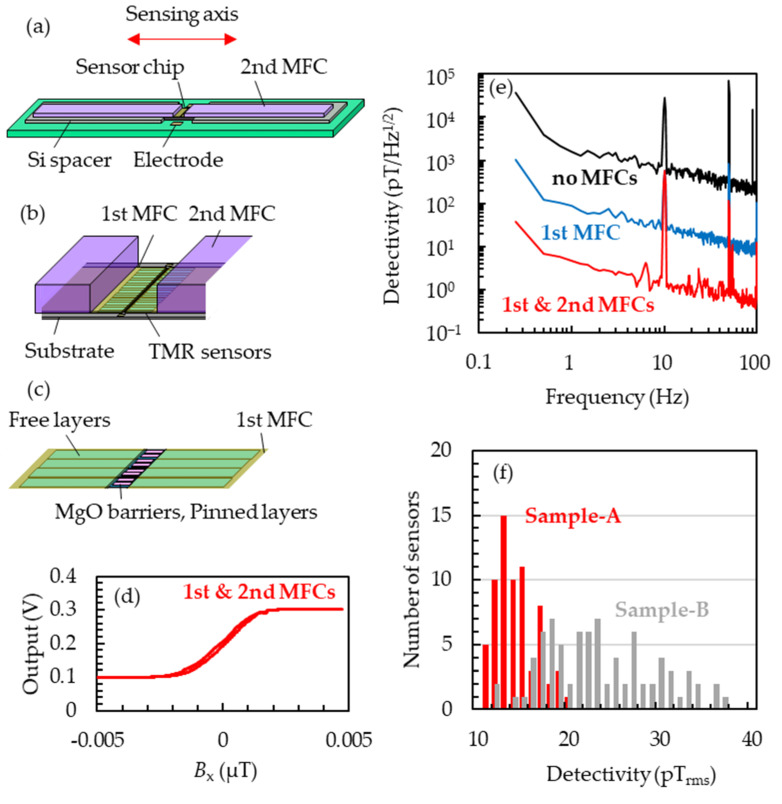
(**a**) Diagram of sensors with MFCs bonded on a print circuit board. The Si spacers and the sensor chip on the board and the second MFCs on the Si spaces are fixed with thermosetting resin. (**b**) Enlarged diagram of the sensor chip in (**a**). The wiring between the sensors and electrodes is omitted for simplicity. (**c**) Enlarged diagram of TMR sensors. (**d**) Transfer curve of a TMR sensor with the first and second MFCs. (**e**) Detectivity spectrum of TMR sensors with/without the first and second MFCs measured in an open-loop system with the AC magnetic field of 1 nT_p-p_ at 10 Hz along the sensing axis. (**f**) Detectivity histogram measured with 108 sensors of Sample-A and 124 sensors of Sample-B. Red and gray data correspond to Sample-A and Sample-B, respectively.

**Figure 2 sensors-23-00646-f002:**
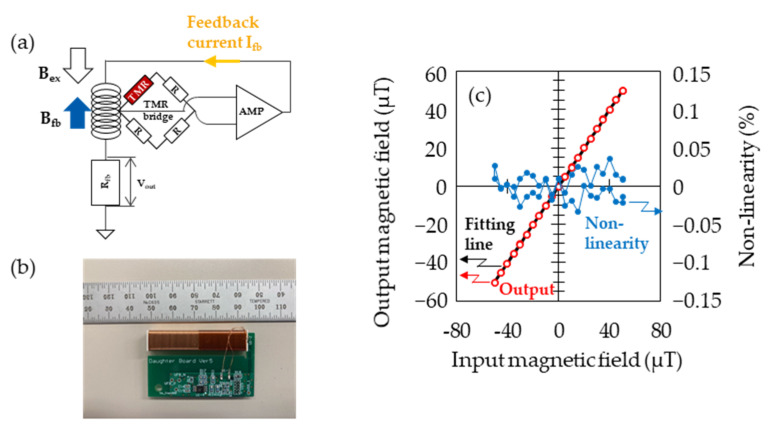
(**a**) Diagram of measurement using the closed-loop system. The amplified differential output from the bridge circuit with a TMR sensor used as a resistive element was induced to the feedback coil. (**b**) Photograph of TMR sensor covered by feedback coil. (**c**) Sensor output and non-linearity in the range of +/−50 μT. Non-linearity was calculated from V_out_ after AD conversion, which was settled within the error of +/−0.1%.

**Figure 3 sensors-23-00646-f003:**
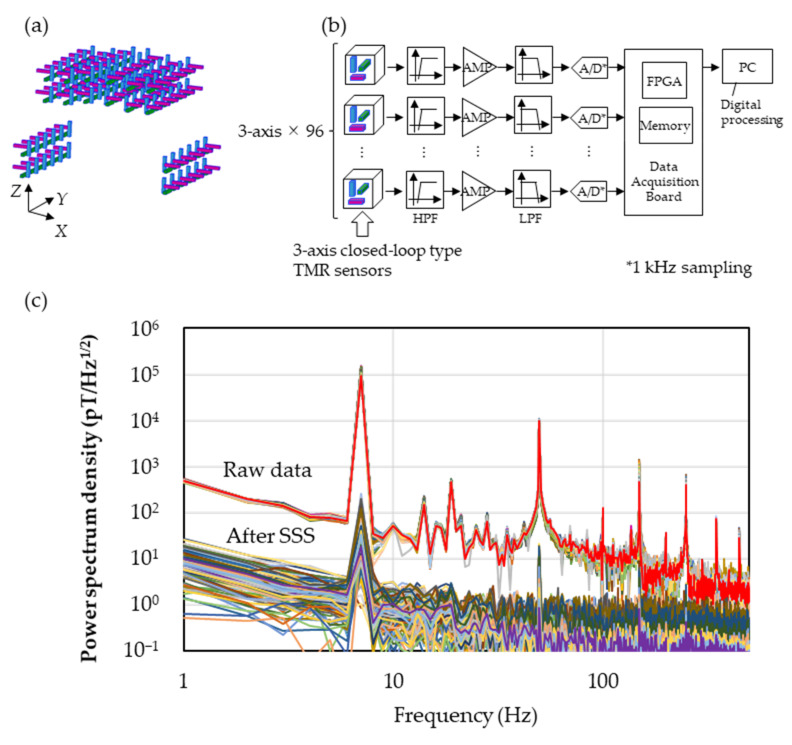
(**a**) Diagram of sensor array. (**b**) Diagram of magnetocardiography measurement. Measured information of three-dimensional magnetic field distribution is stored in the memory via 0.2 Hz high-pass filter, low-noise amplifier, 194 kHz low-pass filter and 24-bit A/D converter. Collected data in memory were sent to a PC through FPGA after being stored for a definite period of time, where digital processing took place. (**c**) Power spectrum density before and after applying SSS measured with the environmental magnetic field at 7 Hz along the *Y*-axis.

**Figure 4 sensors-23-00646-f004:**
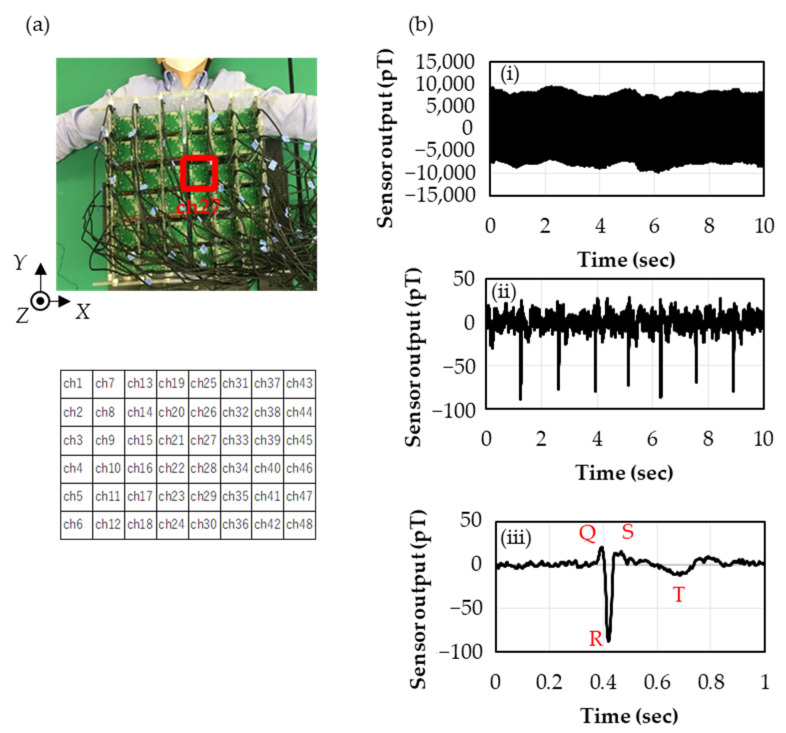
(**a**) Photograph of magnetocardiography measurement. Cells underneath show the channel number at the corresponding position. (**b**) Time series of magnetocardiogram in the digital processing measured by *Z*-sensors in the lower layer in ch27: (i) Raw data (Φ), (ii) After SSS and Digital Filter and (iii) After signal average for one minute (50 heart beats).

**Figure 5 sensors-23-00646-f005:**
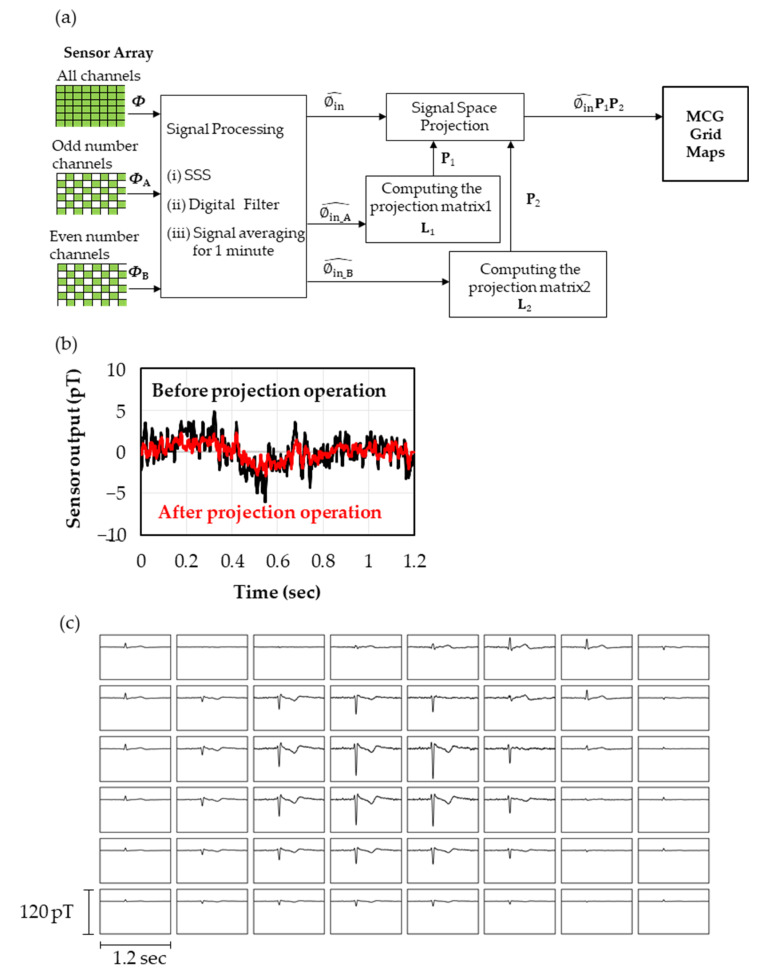
(**a**) Processing flow of projection operation. (**b**) Time series of *Z*-sensors in the lower layer in ch2 measured without the subject to evaluate the change of noises before and after the projection operation. The standard deviation before and after the projection operation was calculated to be 1.77 pT_rms_ and 0.99 pT_rms_, respectively. (**c**) Magnetocardiogram map obtained with the *Z*-sensors in the lower layers in 48 channels after SSS, digital filter, signal average and projection operation.

**Figure 6 sensors-23-00646-f006:**
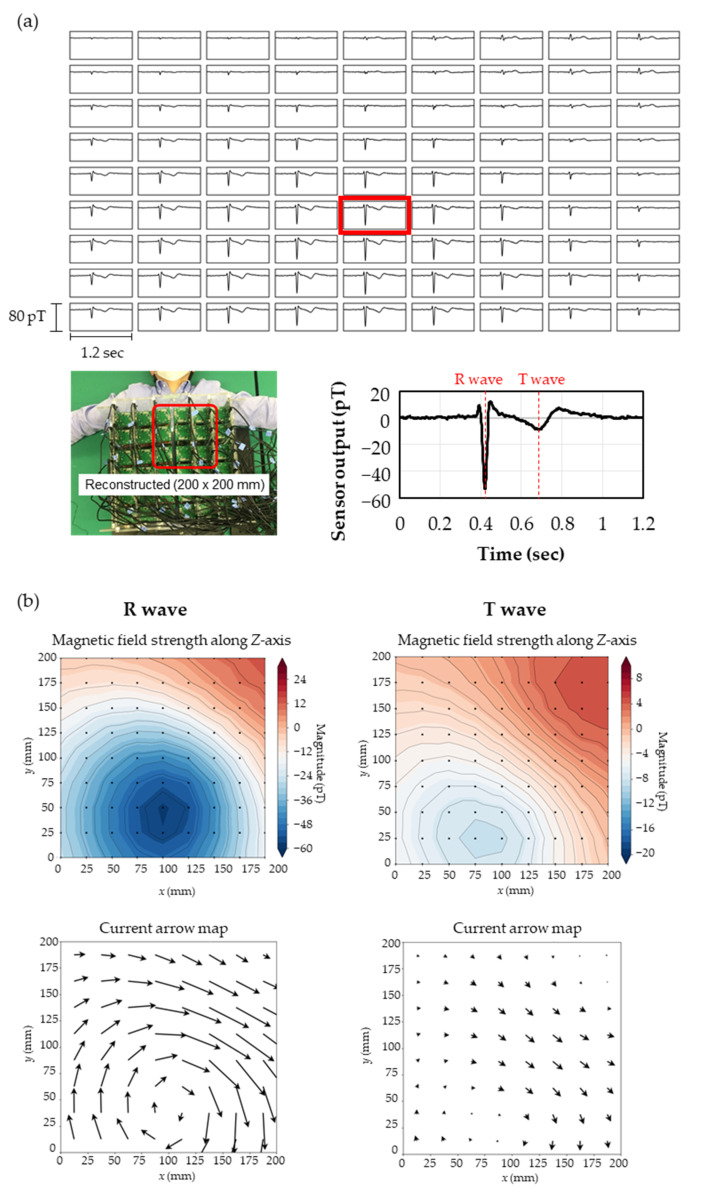
(**a**) Top: magnetocardiogram map reconstructed in the area of 200 × 200 mm at 40 mm above the body surface with 25 mm intervals; lower left: reconstructed area; lower right: expanded magnetocardiograph of data surrounded with the red square on the magnetocardiogram map. (**b**) Magnetic field strength maps in contour lines and current arrow maps at peaks of R and T waves.

**Figure 7 sensors-23-00646-f007:**
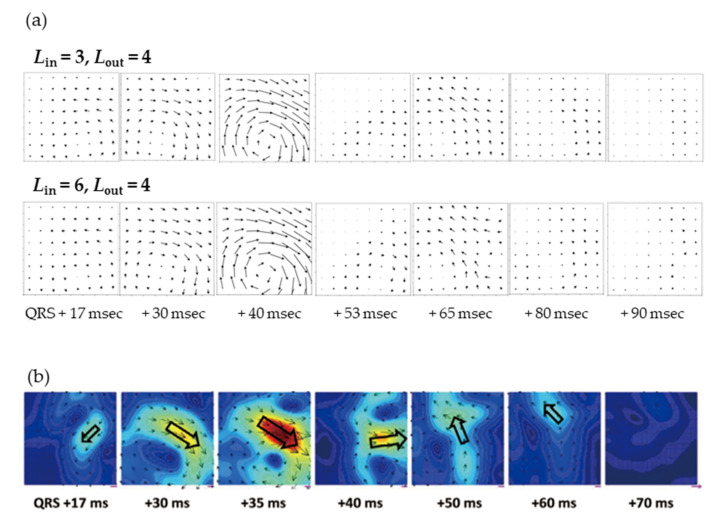
(**a**) Time series of current arrow maps in the QRS domain (300~390 msec in the lower left image in Figure 6a) at each time point referencing the beginning of the QRS wave (300 msec). SSS was performed with *L*_in_ = 6 and *L*_out_ = 4 using sensors in the lower layer of each channel. (**b**) Reference of current arrow maps measured by 64 channels of SQUID sensors with magnetically shielded rooms (Kimura et al., *Circ. J*. 2018).

## Data Availability

Data are available from the corresponding author upon request.

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
