# Peer review of "Development of Magnetocardiograph without Magnetically Shielded Room Using High-Detectivity TMR Sensors"

_sensors, 2023, doi:10.3390/s23020646_

Round 1

Reviewer 1 Report

This paper describes a very interesting simple magnetocardiograph that uses a TMR sensor and signal space separation (SSS) technology.

The TMR sensor uses a two-layer MFC (magnetic flux concentrator) and a feedback circuit to achieve a high sensitivity of 10 pT at 1 Hz. The SSS is also highly effective, significantly reducing environmental noise.I believe that this is a very excellent paper.

However, I expect the following minor changes.

The polarity of the magnetocardiogram waveforms and contour maps in Figures 5 and 6 are opposite. (It may be a good idea to refer to Ann Noninvasive Electrocardiol 2008;13(4):391-400 Standard Template of Adult Magnetocardiogram.

The TMR seosor should have a large 1/f noise, but it would be good to specify how this was resolved.

It would also be good to state what the iintrinsic noise of the TMR seosor is in the calculation and what the noise level is in the magnetic shield.

TMR seosor has MFC and feedback coils, which may cause interference with each other. It would be good to have a description of how this problem is solved.

Author Response

Thank you very much for dedicating to provide your valuable feedback on our manuscript. We are grateful to have your insightful comments. Here, we reply to your comments and suggestion. 

  1. The polarity of the magnetocardiogram waveforms and contour maps in Figures 5 and 6 are opposite. (It may be a good idea to refer to Ann Noninvasive Electrocardiol 2008;13(4):391-400 Standard Template of Adult Magnetocardiogram.)
    (Response)The polarity of waveforms and contour maps in Figures 4, 5, 6 were certainly opposite. We modified these figures.
  2. The TMR seosor should have a large 1/f noise, but it would be good to specify how this was resolved.
    (Response) In terms of the improvement of TMR sensors, we reduced the magnetic 1/f noise by selecting the amorphous Co70.5Fe4.5Si15B10, known to have the excellent flatness, as the material for the free layers as discussed in Section 4.1 and shown in Figure A1. We added a sentence in line 525 to emphasize it. We further decreased the sensor noise by -6.0 dB by adopting the projection operation and SSS, as discussed in Section 4.3.
  3. It would also be good to state what the iintrinsic noise of the TMR seosor is in the calculation and what the noise level is in the magnetic shield.
    (Response) The detectivity of TMR sensors are 4.8 pT/Hz1/2 at 1Hz and 11.8 pTrms in the frequency band between 0.2 and 100 Hz as shown in line 177, and the sensor noise at 1 Hz was 1,714 nV/Hz1/2/V which was performed in a triple magnetically shielding boxes. We added the description of the sensor noise in line 187. The magnetically shielding boxes are generally known to dramatically decrease the environmental noise by about -60 dB as referred in line 135. I hope my explanation is appropriately answering your question.
  4. TMR seosor has MFC and feedback coils, which may cause interference with each other. It would be good to have a description of how this problem is solved.
    (Response) The physical interference between MFCs and feedback coils are avoided by mechanically fixing the coils on a printed circuit board. We added a description in line 141. In terms of the magnetic interference, the magnetic field generated by coils are nicely coupled to the MFCs and TMR sensors, namely, the external fields are well cancelled by the feedback coils as described in line 197.

Reviewer 2 Report

This paper introduced a method to detect magnetocardiograph by using TMR sensors. Firstly, the author used Co70.5Fe4.5Si15B10 as the free layer and the TMR sensor with detectivity of 14.1 pTrms on average in the frequency band between 0.2 ~ 100 Hz was prepared by using MFC. Then the SSS was used to shield the ambient noise by mathematical method, and the measurement of magnetocardiograph in unshielded environment is realized. The results obtained by this method arecomparing favorably with a measurement by a conventional SQUID performed in magnetically shielded rooms.

In generally, the article is of high quality and may attract broad readers. I suggest this article to be published with a few changes. Minor comments are there are some small mistakes in the article, e.g., ‘-296℃’. And I have a few more questions which I hope the author can answer

1     The author used an innovative design of sensors free layer in the manuscript, using free layer as a bottom electrode can increase the free layer volume and result in the reduction in sensors detectivity. However, I was wondering if only the induced anisotropy field in the first annealing is enough to linearize the sensor. Can the author provide the transfer curve or magnetic characterization data of the sensor?

2     Can author explain why use Co70.5Fe4.5Si15B10 100/Ru 10/Co70.5Fe4.5Si15B10 100/Ru 0.4/Co40Fe40B20 3 (nm) as free layer? (What is the function of Co70.5Fe4.5Si15B10 100/Ru 10 ?)

Author Response

Thank you very much for dedicating to provide your valuable feedback on our manuscript. We are grateful to have your insightful comments. Here, we reply to your comments and suggestion.  

  1. The author used an innovative design of sensors free layer in the manuscript, using free layer as a bottom electrode can increase the free layer volume and result in the reduction in sensors detectivity. However, I was wondering if only the induced anisotropy field in the first annealing is enough to linearize the sensor. Can the author provide the transfer curve or magnetic characterization data of the sensor?
    (Response) The second annealing was carried out to rotate the pinned layer by 90 degrees. By orthogonalizing the magnetic easy axes of the pinned and the free layers, the transfer curve with the magnetic field along the hard axis of the free layers reflects the M-H curve for hard direction of the free layers. As you suggested, adding a figure of the transfer curve will give further information to the readers. We modified the description in line 129 and added a sentence in line 177, and added a figure of the transfer curve of Sample-A with the first and second MFCs as Figure 1d.
  2. Can author explain why use Co70.5Fe4.5Si15B10 100/Ru 10/Co70.5Fe4.5Si15B10 100/Ru 0.4/Co40Fe40B20 3 (nm) as free layer? (What is the function of Co70.5Fe4.5Si15B10 100/Ru 10 ?)
    (Response) Since Co70.5Fe4.5Si15B10 films have soft magnetic properties and are strongly coupled to Co40Fe40B films, the small anisotropy field is achieved (Appl. Phys. Exp. 2013, 6, 103004). The double Co70.5Fe4.5Si15B10 films are necessary to eliminate the magnetic domain structure. Two Co70.5Fe4.5Si15B10 films separated by the Ru layer magnetostatically couple on the side of the pattern, which leads to the single magnetic domain. We hope you will find the discussion in line 432.
